# Role of Self-Sampling for Cervical Cancer Screening: Diagnostic Test Properties of Three Tests for the Diagnosis of HPV in Rural Communities of Cuenca, Ecuador

**DOI:** 10.3390/ijerph19084619

**Published:** 2022-04-12

**Authors:** Bernardo Vega Crespo, Vivian Alejandra Neira, José Ortíz Segarra, Ruth Maldonado Rengel, Diana López, María Paz Orellana, Andrea Gómez, María José Vicuña, Jorge Mejía, Ina Benoy, Tesifón Parrón Carreño, Veronique Verhoeven

**Affiliations:** 1Facultad de Ciencias Médicas, Universidad de Cuenca, Cuenca 010203, Ecuador or vneira@uazuay.edu.ec (V.A.N.); jose.ortiz@ucuenca.edu.ec (J.O.S.); pazorellanajara@gmail.com (M.P.O.); angiegomeza@gmail.com (A.G.); joshevicuna@hotmail.com (M.J.V.); jorge.mejia@ucuenca.edu.ec (J.M.); 2Facultad de Medicina, Universidad del Azuay UDA, Cuenca 010104, Ecuador; dilopez@uazuay.edu.ec; 3Facultad de Ciencias de la Salud, Universidad Técnica Particular de Loja UTPL Loja Ecuador, Loja 1101608, Ecuador; remaldonado6@utpl.edu.ec; 4Programa de Doctorado en Ciencias Morfológicas, Universidad de La Frontera UFRO, Temuco 4811230, Chile; 5AMBIOR, Laboratory for Cell Biology & Histology, University of Antwerp, 2610 Antwerp, Belgium; ibenoy@ambior.org; 6Facultad de Ciencias de la Salud y Neurociencias, Universidad de Almería UAL, 04120 Almería, Spain; tpc468@ual.es; 7Family Medicine and Population Health, University of Antwerp, 2610 Antwerp, Belgium; veronique.verhoeven@uantwerpen.be

**Keywords:** HPV, self-sampling, urine sampling, clinician sampling, sensitivity and specificity

## Abstract

Background: HPV primary screening has shown effectiveness for cancer prevention; however, gynaecological examination is considered uncomfortable. Self-sampling methods increase the acceptance of screening. The aim of this study is to compare the sensitivity and specificity of clinician sampling versus vaginal and urine self-sampling for HPV diagnosis. Methods: A diagnostic test study was conducted in a rural parish of Cuenca, Ecuador. A total of 120 women participated. Each participant self-collected urine and vaginal samples and underwent clinician sampling for HPV testing. The latter was considered as the golden standard. All three samples were processed with the same amplification and hybridization protocol for HPV detection (Hybribio) following the manufacturer’s instructions. Results: Characteristics of the participants were: median age 35 years; 40.8% married; 46.7% had a primary level of education; and median age of sexual onset, 17.6 years. The prevalence of any type of HPV with clinician sampling was 15.0%, 17.5% with urine sampling and 18.3% with vaginal self-sampling. Self-sampling sensitivity reached 94.4% (IC 74.2–99.9), and specificity 92.1% (IC 85.2–95.9). Urine sampling had a sensitivity of 88.8% (IC 67.2, 96.9), and specificity 94.1% (IC 67.2–96.9). The negative predictive value was 98.9% (IC 94.2–99.8) for vaginal self-sampling and 97.6% (IC 92.6–99.4) for urine sampling. Conclusions: This study shows that vaginal and urine self-sampling methods have similar sensitivity and specificity compared with clinician sampling for the diagnosis of HPV. The correlation between HPV genotypes among the three tests is satisfactory.

## 1. Introduction

Since the pap smear was implemented for population screening in 1970 for cervical cancer (CC) prevention, mortality has decreased by 70% worldwide [1]. However, the sensitivity of the conventional pap smear for the detection of cellular abnormalities varies between 51% and 55%, and the specificity is between 66.6% and 75% [2,3]. Due to those rates, the success of this test depends on the adherence to repeated screening during a woman’s lifetime in order to find precancerous lesions in early stages. To reach screening objectives, it is recommended that the intervals between pap smears last no longer than 3 years [4].

The detection of human papilloma virus (HPV) used as a primary screening method has a higher sensitivity (95%) and specificity (94% to 95%) than cytology to detect premalignant lesions [5,6,7]. A negative HPV test could extend the intervals of screening to five years according to some protocols [5,8]. The global strategy for cervical cancer prevention of the World Health Organization (WHO) considers that CC mortality could effectively be prevented if every woman has at least two high-sensitivity cervical screening episodes during their lifetime (at 35 and 45 years old) [9].

Screening and vaccination against HPV are reducing CC mortality in countries with strong preventative programs [10]. However, in low- and middle-income countries (LMIC), barriers to cervical cancer screening persist [11]. The rates of under-screened women are variable among countries. In Ecuador, 41.6% of women of reproductive age have never been screened [12].

Several barriers have been identified for access to cervical CC screening services; according to the socio-ecological model, they interact at different levels: organizational (difficulties in access to health centres and long waiting times), interpersonal (stigma and lack of family support) and personal (lack of risk perception and fear of examination) [13,14,15]. Furthermore, the COVID-19 pandemic has increased the disparities in access to healthcare facilities [16].

Self-sampling methods, such as vaginal self-sampling and urine sampling for HPV diagnosis, have demonstrated high acceptability and sensitivity for cervical cancer screening [17,18]. Compared with clinician sampling, they are less invasive and could be more attractive to increase the uptake in under-screened and never-screened women and to overcome barriers at different levels of interaction [18,19]. 

On the other hand, doubts about the sensitivity of self-sampling tests could discourage women to choose those tests. There are limited studies that compare the sensitivity and specificity of vaginal self-sampling and urine sampling for HPV detection at the same time and in the same population. 

A literature review reflects a wide range of variation in sensitivity and specificity among tests and techniques [20,21,22,23,24,25,26].
**Type of Sample****Sensitivity****Specificity****Author**Vaginal self-sampling83.3%73.9%Asciutto et al., 2018Vaginal self-sampling84%93%Arbyn et al., 2018Vaginal self-sampling84.6%62.9%Wang et al., 2020Vaginal self-sampling50%98% Esber et al., 2018Vaginal self-sampling98.9%100%Kuriakos et al., 2019Urine self-sampling48.1%82.8%Asciutto et al., 2018Urine self-sampling90.5%74.0%Combita et al., 2016

This is the first study of self-sampling effectiveness conducted in Ecuador; the aim of this research is to compare sensitivity, specificity, predictive values and correlation for the diagnosis of HPV from vaginal self-sampling and urine self-sampling versus clinician sampling in a rural setting.

## 2. Materials and Methods

### 2.1. Ethical Statement 

This study was approved under the guidance of the Declaration of Helsinki and the Council for International Organizations of Medical Sciences (CIOMS). All procedures involving human participants were approved by the bioethical committee of the University of Cuenca (approval code UC-COBIAS-2020-262) and the National Directory of Intelligence in Health (DIS) of the Ministry of Health of Ecuador (approval code MSP-DIS-2020-0405-O). All the participants were informed about the purpose of the study and signed an informed consent form before the sample collection.

### 2.2. Study Population

A diagnostic test study was conducted in the rural parish of El Valle of Cuenca Ecuador from May to August 2021. Through flyers delivered in public places, households and in healthcare centres, women of El Valle were invited to participate. 

The inclusion criteria included being sexually active; being between 18 and 70 years old; not having undergone an excision or destructive treatment of the cervical intraepithelial neoplasm; not having used vaginal medication at least a week before the examination; not having had sexual intercourse for at least 48 h previous to the examination; not being pregnant; and the absence of menstrual bleeding at the time of examination. One hundred twenty women participated and provided three samples: urine, a vaginal sample and a physician-taken cervical assay. 

### 2.3. Sampling Collection

Prior to sample collection, patients who agreed to participate were taught how to take the samples: a pictographic representation of each technique was given to all participants and the same graphics were present in the bathroom where the patients were instructed to obtain the samples. 

The first sample obtained was a urine sample; it was obtained directly in a sterile urine container after self-made asepsis by the patient in the bathroom of the consultation room. All patients were asked to collect at least 30 cc of urine. 

After urine sampling, a self-sampling device for vaginal sampling was given to the patient. An Evalyn^®^ Brush from Rovers Medical Devices was the selected tool for this sampling. Fabricant instructions were used for sample collection; researchers waited outside of the bathroom for any additional explanation and to receive the sample after its collection. Finally, the patient was directed to a gynaecological examination table. After the insertion of the speculum, endocervix and exocervix samples were obtained by using a Hybribio cervical brush, rotating 360° twice. The cervical brush was placed in Roche Cell Collection Medium for transportation. This medium was selected because it contains 20 cc of preservant, which allows the researcher to centrifuge and obtain material for HPV diagnosis.

All the samples were paired, labelled, coded and transported to the laboratory of molecular biology of the University of Cuenca within the first 6 h of collection. The results for HPV were delivered to the patients within 10 days after the sample collection. 

### 2.4. HPV Genotyping 

Once in the laboratory, the 3 samples (clinician sampling, self-sampling, and urine sampling) from each patient were processed to obtain the genetic material. For urine DNA purification, we used the DNA prep kit (Hybribio, Guangzhou, China) following the manufacturer’s instructions. For the conventional collection samples, we used the Cell Lysis Kit (Hybribio) following the manufacturer’s protocol. Finally, the self-sampling brushes were washed for about 1 min in Hybribio Female Sample Collection media to release the cells and purify the genetic material using the same extraction kit as the conventional collection. All the material obtained was stored at −20 °C until further use. 

For amplification, we used the 37 HPV GENOARRAY kit (Hybribio), which allows the simultaneous amplification of 37 different genotypes, including high-risk genotypes: 16, 18, 31, 33, 35, 39, 45, 51, 52, 53, 56, 58, 59, 66, and 68; and the low-risk (LR) ones: 6, 11, 42, 43, 44, and CP8304 (81); and 26, 34, 40, 54, 55, 57, 61, 67, 69, 70, 71, 72, 73, 82, 83, 84 categorized as undetermined risk. The PCR mix was performed according to the manufacturer’s instructions to obtain a final reaction volume of 25 μL (23.25 μL PCR Mix, 0.75 μL of DNA Taq polymerase 5 U/μL and 1 μL of DNA). For urine samples, the final volume was 26 μL because we added 2 μL of DNA template. Amplification was performed in the Veriti Thermal Cycler (Applied Biosystems, Waltham, MA, USA) with the following programming: initial denaturation at 95 °C for 5 min, 40 cycles of denaturation at 95 °C for 20 s, annealing at 55 °C for 30 s and elongation at 72 °C for 30 s, to finish with a final elongation at 72 °C for 5 min.

Finally, all the amplicons were denatured for 5 min at 95 °C and placed on ice before continuing the hybridization. The process was carried out in the HibriMax (Hybribio) according to the manufacturer’s instructions. We used HPV-37 Hybrimem membranes containing the immobilized probes of the genotypes of interest. Streptavidin–horseradish peroxidase conjugate was added to bind to the biotinylated PCR products. The direct visualization of the breakdown product (purple precipitate) of the substrate nitroblue tetrazolium-5-bromo-4-chloro3-indolylphosphate was interpreted as positive for the corresponding HPV DNA type as indicated on the schematic diagram of the membrane provided with the test kit.

### 2.5. Data Analysis 

Completed questionaries with sociodemographic data and the results of HPV tests were transcribed to a Microsoft Excel 2016 spreadsheet for cleaning and coding and were subsequently transferred to the Statistical Package for the Social Sciences for Windows version 17.0 (IBM, Armonk, NY, USA). Descriptive statistics were presented using means and standard deviation (SD) for continuous variables and percentages for categorical variables. Open-Source Epidemiologic Statistics for Public Health (Rollins School of Public Health de la Universidad de Emory, Atlanta, GA, USA) was used to calculate sensitivity, specificity, positive predictive value, negative predictive value, likelihood positive ratio, likelihood and Cohen’s kappa. The kappa statistic was calculated to determine the level of chance agreement between the two methods, with a kappa value of 0 indicating no agreement better than chance, 1 indicating perfect agreement better than chance, and intermediate values of 0.00–0.20, 0.21–0.40, 0.41–0.60, 0.61–0.80 and >0.81 indicating poor, fair, moderate, good and excellent agreement. 

Clinician cervical sampling with a speculum is the standard method for the diagnosis of HPV; therefore, the sensitivity and specificity of HPV detection in urine and vaginal self-sample were calculated using cervical sampling as reference. The width of the confidence intervals of kappa and the diagnostic accuracy parameters show the precision of our estimates.

## 3. Results

### 3.1. Population Characteristics

A total of 120 women participated in this study, all of them living in the rural parish. The sociodemographic characteristics of the participants are shown in Table 1. 

### 3.2. Comparison of Tests

The prevalence of any type of HPV with clinician sampling was 15.0%, 17.5% with urine self-sampling and 18.3% with vaginal self-sampling.

Table 2 presents the most prevalent HPV genotypes found in the three sampling techniques: 58, 51, 31, 52, 53, and 16. The following genotypes were detected in self-sampling but not in clinician sampling: 11, 33, 68, and 72. On the other hand, 11, 54, 68, and 73 were detected in urine sampling and not in clinician sampling.

Table 3 shows the comparison among the tests: the self-sampling sensitivity reached 94.4% (IC 74.2–99); specificity, 92.1% (IC 85.2–95.9); predictive positive value (PPV), 68.0% (IC 48.4–82.8); predictive negative value (PNV), 98.9% (IC 94.28, 99.81); positive likelihood ratio (PLR), 12 (IC 9.36–15.49); and negative likelihood ratio (NLR), 0.06 (IC 0.008–0.428). Agreement with clinician sampling was 0.74 (kappa).

The urine sampling had a sensitivity of 88.8% (IC 67.2, 96.9); specificity of 94.1% (IC 87.76, 97.28); PPV of 72.2% (IC 51.85, 86.85); PNV of 97.6% (IC 51.85–86.85); PLR of 15 (IC 10.73–21.27); and NLR 0.11 (IC 0.04–0.315). The agreement with clinician sampling was 0.76 (kappa).

## 4. Discussion

This study demonstrates that urine self-sampling and vaginal self-sampling have similar sensitivity and specificity to clinician sampling for the diagnosis of HPV.

The general prevalence of HR and LR HPV genotypes is highly variable in the literature. In Europe, the range of HPV positivity goes from 2% in Spain to 12% in Belgium [27]. In Ecuador, this variation is also present. Cabrera J. et al. in 2015 reported a prevalence of 25.6% in Cuenca, in the Azuay province of Ecuador [28]. González-Andrade F. et al. in 2019 reported a prevalence of 6.3% among mestizo women [29]. Our findings (15%) are similar to the results presented by Dunne E. et al. in the United States, showing a prevalence of 17% [30]. A possible explanation for the high prevalence of HPV in a rural population is the large number of women below 30 years included in the study [31].

The prevalence of any type of HPV was slightly higher in vaginal self-sampling (18.3%) and urine sampling (17.5%); similar results were found by Polman N. et al. in 2019 in the Netherlands, where the HPV positivity rate with self-sampling was 7.4% vs. 7.2% with clinician sampling [32]. In urine and vaginal self-sampling, a higher number of cells could be recovered from the genital tract when compared with clinician sampling, which only collects endo- and exocervical cells. That could explain the higher rates of positivity in self-sampling methods.

HPV genotype also presents variation worldwide. In Europe, the most prevalent genotypes are 16, 18, 31, and 33 [27], and in China, they are 52, 58, 31, 52, 39, and 68 [31]. Our study found similar results to those presented by González-Andrade F in Ecuador: 16, 18, 31, 52, 53, 56, and 58. This variation could be explained by the epidemiological prevalence of HPV according to the area where the participants involved in each study reside [29].

The values of the sensitivity and specificity of vaginal self-sampling are comparable to the findings of Arbyn et al., 2018 (84% and 93%) [21], and are lower than the results from Kuriakos et al., 2019 (98.9% and 100%) (26). The standardization of the technique could explain this variation. However, this method has demonstrated equal efficiency to clinician sampling in large studies [32].

For urine sampling, our results are similar to the sensitivity and specificity reported by Combita et al. in 2016 (90.5–74.0%) [22]. However, our research sensitivity was higher. This difference could also be explained by the employed technique: this research used a reactive substance designed specifically for urine, which could increase the effectiveness.

A pap smear has a mean sensitivity of 51% [2]. Both methods, urine and vaginal self-sampling, could be considered more efficient because their sensitivity is higher than 80% [33]. Screening would be equally reliable and could be less frequent than pap smears. In addition, the high specificity and negative predictive value are relevant for clinical practice, because patients with negative results rarely present a cervical lesion and therefore have low risk for cervical cancer [6,34].

Our research reported a good kappa correlation with clinician sampling; similar results are reported by Swift et al., 2020, with agreement reaching 0.73 [35]. This situation reinforces the effectiveness of self-sampling methods for the primary screening of HPV.

Self-sampling methods (urine and vaginal sampling) have a high acceptability among populations in rural areas [36], which could increase the uptake of examination. In addition, self-samples may be collected in low-infrastructure settings and be offered by midwives and healthcare workers at the community level [37].

### Limitations

A limitation of the study is that participants were selected by convenience, and all patients who agreed to participate and met the study criteria were included in the study. However, women that participated had similar sociodemographic characteristics, making them comparable. Another limitation is that an important number of women below 30 years old participated in the research, which could increase the prevalence rates of HPV in this study. This affects PPV and NPV in our sample; typically, in populations with a lower prevalence, NPV would be better.

Including young women was intentional in order to find more HPV-positive patients and to have a sufficient number of positive results to calculate sensitivity and specificity. Previous research conducted in Ecuador demonstrated a lower prevalence of HPV in rural areas compared with urban.

## 5. Conclusions

This research study is the first conducted in Ecuador evaluating the effectiveness of self-sampling methods in a rural community in Cuenca, Ecuador.

This study shows that vaginal and urine self-sampling methods have similar sensitivity and specificity to clinician sampling for the diagnosis of HPV. The Kappa correlation between HPV genotype among the three tests is good.

Self-sampling methods have a high diagnostic capacity and a capability for the detection of positive cases of HR HPV.

The positive results of LR HPV and absence of HPV have a high relevance in clinical practice for detecting or discarding risk for cervical cancer.

Self-sampling methods have high sensitivity and specificity and have demonstrated reliability in rural settings.

## Figures and Tables

**Table 1 ijerph-19-04619-t001:** Participants’ socio-demographic characteristics.

Variable	N (%)
Age: mean 35; mode 24; SD ± 11.23	
19 to 29	42 (35.5)
30 to 39	32 (26.7)
40 to 49	31 (25.8)
50 to 59	12 (10.8)
60 to 69	2 (1.7)
Educational level	
None	8 (6.7)
Alphabetization centre	1 (0.8)
Primary School	56 (46.7)
High school	43 (35.8)
University	11 (9.2)
Post graduate	1 (0.8)
Civil status	
Married	49 (40.8)
Living as a couple	28 (23.3)
Single	25 (20.8)
Divorced	11 (9.2)
Separated	3 (2.5)
Widow	4 (3.3)
Occupation	
Housewife	69 (57.5)
Employed	27 (22.5)
Agriculture	3 (2.5)
Student	2 (1.7)
Retired	1 (0.8)
Stylist	1 (0.8)
Seller	1 (0.8)
Cleaning	1 (0.8)
Others	3 (2.5)
Family Income (USD)	
<100	22 (18.3)
100 to 200	21 (17.5)
201 to 300	19 (15.8)
301 to 400	23 (19.2)
401 to 500	17 (14.2)
501 to 600	6 (5.0)
>600	12 (10.0)
Age of sexual onset: median 17.6; mode 18; SD ± 2.9	
9 to 14 years	12 (10.0)
15 to 19 years	82 (68.3)
20 to 24 years	23 (19.2)
25 to 29 years	2 (1.7)
30 to 34 years	1 (0.8)
Previous cervical screening	
Yes	98 (81.7)
No	22 (18.3)

**Table 2 ijerph-19-04619-t002:** Distribution of any type of HPV according to sampling method.

Genotype	11 -	16 *	18 *	31 *	33 *	39 *	51 *	52 *	53 **	54-	56 *	58 *	66 **	68 *	70 -	71 -	72 -	73 *	81 -	84 -
Clinician sampling	-	1 (4.3)	1 (4.3)	3 (13.0)	-	1 (4.3)	3 (13.0)	2 (8.7)	2 (8.7)	-	1 (4.3)	4 (17.4)	1 (4.3)	-	1 (4.3)	1 (4.3)	-	-	1 (4.3)	1 (4.3)
Self-sampling	1 (3.2)	2 (6.5)	1 (6.5)	3 (9.7)	1 (3.2)	1 (3.2)	4 (12.9)	4 (12.9)	2 (6.5)	-	1 (3.2)	5 (19.4)	1 (3.2)	1 (3.2)	1 (3.2)	1 (3.2)	1 (3.2)	-	-	1 (3.2)
Urine sampling	1 (3.4)	2 (6.9)	1 (6.9)	3 (10.3)	-	1 (3.4)	4 (13.8)	1 (3.4)	2 (6.9)	2 (6.9)	1 (3.4)	4 (13.8)	2 (6.9)	1 (3.4)	1 (3.4)	-	-	1 (3.4)	1 (3.4)	1 (3.4)

* High-risk HPV. ** Middle-risk HPV. - Low-risk HPV.

**Table 3 ijerph-19-04619-t003:** Comparison of sensitivity, specificity, predictive values, likelihood ratio and correlation among three tests.

		Clinician Sampling		Sensitivity	Specificity	PPV	PNV	LR+	LR−	Kappa
	Result	Positive *n* (%)	Negative *n* (%)	% (IC 95%)	% (IC 95%)	% (IC 95%)	% (IC 95%)	*n* (IC)	*n* (IC)	*n* (IC)
Self-sampling	Positive	17 (14.2)	8 (6.7)	94.4 (74.2–99.0)	92.1 (85.2–95.9)	68.0 (48.4–82.8)	98.9 (94.28, 99.81)	12.0 (9.361–15.49)	0.06 (0.008–0.428)	0.74 (0.57–0.92)
	Negative	1 (0.8)	94 (78.3)							
Urine sampling	Positive	16 (13.3)	6 (5.0)	88.8 (67.2, 96.9)	94.1 (87.76, 97.28)	72.7 (51.85, 86.85)	97.6 (92.86, 99.44)	15.1 (10.73–21.27)	0.11 (0.044–0.315)	0.76 (0.58–0.93)
	Negative	2 (1.7)	96 (80.0)							

## Data Availability

The datasets generated and/or analysed during the current study are not publicly available because they contain the sensitive personal information of participants. The informed consent grants the confidentiality of the participants’ data. However, the datasets are available from the corresponding author upon reasonable request.

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
