# Peer review of "Role of Self-Sampling for Cervical Cancer Screening: Diagnostic Test Properties of Three Tests for the Diagnosis of HPV in Rural Communities of Cuenca, Ecuador"

_ijerph, 2022, doi:10.3390/ijerph19084619_

Round 1

Reviewer 1 Report

HPV self-sampling can potentially be a good way to increase cervical screening uptake in Ecuador and it is important that local studies are conducted to evaluate this.  This type of comparisons have been previously published, so the idea is not entirely new but this study serves to provide some local data.

The methodology is straight forward.  However, the sample size calculations and the statistical analysis need more details to convince the readers that the study is scientifically sound.  ( eg what can this sample size detect ? a difference of ? in sensitivity  ) .  Furthermore, the urine sample sensitivity used in the sample size calculation (48.1% )quoted from Asciutto et al may not be appropriate since Asciutto's study was on detection of HPVmRNA , not DNA

There are many spelling mistakes, giving the impression of a rather careless submission. 

Author Response

Question / Comments  

HPV self-sampling can potentially be a good way to increase cervical screening uptake in Ecuador and it is important that local studies are conducted to evaluate this.  This type of comparisons have been previously published, so the idea is not entirely new but this study serves to provide some local data.

Response: Thank you so much for your valuable comments. Indeed, there are several studies that compare self sampling versus clinician sampling. The advantage of this study is that samples are collected at the same time in the same population, and that we compare 3 tests. Another advantage is that was tested for the first time in Ecuador and in rural population.  Having local data of sensitivity and specificity available could help policy makers to implement self-sampling tests in a country where 4 of each 10 women haven’t had a screening during life time and also in remote areas.  

Question / Comments  

The methodology is straight forward.  However, the sample size calculations and the statistical analysis need more details to convince the readers that the study is scientifically sound.  ( eg what can this sample size detect ? a difference of ? in sensitivity  ) .  Furthermore, the urine sample sensitivity used in the sample size calculation (48.1% ) quoted from Asciutto et al may not be appropriate since Asciutto's study was on detection of HPVmRNA , not DNA

Response:

Thank you very much for this comment. After seeking statistical advice, we now realise that we used the sample size calculator in a wrong way. We calculated a sample size for a pairwise comparison of self sampling and urine sampling (in EpiDat versión 4.2), while our primary goal was to examine the concordance of each test separately with the golden standard. Thus our calculation was flawed. We ended up with comparing tests in 120 women, which luckily is in line with sample sizes in other studies. We changed this in the manuscript: Thus we do not confirm or reject a null hypothesis in this study, but we describe the diagnostic properties of self sampling and urine testing as compared to the golden standard (as is common in diagnostic studies). The width of the confidence intervals for kappa, sensitivity, specificity, PPV, PPN, LR+ and LR- shows the precision of our estimates. Thank you for pointing out this mistake.

Question / Comments  

There are many spelling mistakes, giving the impression of a rather careless submission.

Response: We have improved quality check using a text editing system

Reviewer 2 Report

The authors presented study titled: " Role of self-sampling for cervical cancer screening: Diagnostic test properties of three tests for the diagnosis of HPV in rural communities of Cuenca, Ecuador." submitted to IJERPH. Paper presented by authors showed very interesting and important topic related to role of self-sampling for cervical cancer screening. Authors demonstrated that vaginal and urine self-sampling methods has similar sensitivity and specificity compared to clinician sampling for the diagnosis of HPV.  According authors correlation between HVP genotype among the three tests is satisfactory. It must be emphasized that this is the first study of self-sampling effectiveness conducted in Ecuador; Additionally authors proved, based on comparison of sensitivity and specificity, that predictive values and correlation for the diagnosis of HPV from vaginal self-sampling, urine self-sampling is very high versus clinician sampling in a rural setting. The main criticism is that sampling group is limited to Ecuador patients and in my opinion it should be extended to another countries within scientific collaborations, then results should be more representative.

Author Response

Question / Comments  

The authors presented study titled: " Role of self-sampling for cervical cancer screening: Diagnostic test properties of three tests for the diagnosis of HPV in rural communities of Cuenca, Ecuador." submitted to IJERPH. Paper presented by authors showed very interesting and important topic related to role of self-sampling for cervical cancer screening.

Response Thank you so much for your valuable comments

Question / Comments  

Authors demonstrated that vaginal and urine self-sampling methods has similar sensitivity and specificity compared to clinician sampling for the diagnosis of HPV.  According authors correlation between HVP genotype among the three tests is satisfactory. It must be emphasized that this is the first study of self-sampling effectiveness conducted in Ecuador;

Response. Thank you so much we have emphasized this in several parts of the paper 

Question / Comments   

Additionally authors proved, based on comparison of sensitivity and specificity, that predictive values and correlation for the diagnosis of HPV from vaginal self-sampling, urine self-sampling is very high versus clinician sampling in a rural setting. The main criticism is that sampling group is limited to Ecuador patients and in my opinion it should be extended to another countries within scientific collaborations, then results should be more representative.

Response: Thank you so much. Indeed, we plan to explore funds to promote a collaborative project in other countries 

Reviewer 3 Report

This manuscript was about self-sampling technique of HPV as screening for cancer. I found the manuscript well prepared, easy to understand, and interesting. However, some points must be addressed before publication.

Major points

  1. The authors mentioned sample collection for cytology in line 126 and further colposcopy test for HR-positive participants in line 130. The results of cytology should be shown at least to check how it matches the HPV test.
  2. The acceptability of the self-sampling method was not the focus of this study so the line 271 in the discussion is not needed. Some other lines in the conclusions is also not the conclusion of this study.

Minor point

The word HVP come out many times in the manuscript which seems to mean HPV.

Author Response

Question / Comments  

  1. The authors mentioned sample collection for cytology in line 126 and further colposcopy test for HR-positive participants in line 130. The results of cytology should be shown at least to check how it matches the HPV test.

Response Thank you so much, for your comments, indeed clinician sampling was analysed with molecular biology and cytology. However, as the main objective of this study was to compare HPV presence, this study only focused in that diagnostic test for HPV. 

In addition, cytology could give normal results even if HPV HR or LR are present due the low sensitivity of pap smear and because cellular changes could occur time after HPV infection

We plan to present another article stablishing in which we describe HVP, cytology colposcopy and biopsy in the cases referred to colposcopy

We have retired this paragraph of the article to avoid confusions.    

Question / Comments

  1. The acceptability of the self-sampling method was not the focus of this study so the line 271 in the discussion is not needed. Some other lines in the conclusions is also not the conclusion of this study.

Response: We have retired line 271 conclusions about acceptability   

Question / Comments  

The word HVP come out many times in the manuscript which seems to mean HPV.

Response Thank you so much we have corrected this misspelling We have improved quality check using a text editing system